# Biological complexity facilitates tuning of the neuronal parameter space

**Marius Schneider** [1,2,3,4]*, **Alexander D. Bird**[1,2,3]*, **Albert Gidon**[5], **Jochen Triesch**[1,4,6], **Peter Jedlicka**[3,7⦿], **Hermann Cuntz**[1,2,3⦿]*

**1** Frankfurt Institute for Advanced Studies, Frankfurt am Main, Germany, **2** Ernst Strüngmann Institute (ESI) for Neuroscience in cooperation with the Max Planck Society, Frankfurt am Main, Germany, **3** ICAR3R—Interdisciplinary Centre for 3Rs in Animal Research, Justus Liebig University Giessen, Giessen, Germany, **4** Faculty of Physics, Goethe University, Frankfurt/Main, Frankfurt am Main, Germany, **5** Institute for Biology, Humboldt-Universität zu Berlin, Berlin, Germany, **6** Faculty of Computer Science and Mathematics, Goethe University, Frankfurt am Main, Germany, **7** Institute of Clinical Neuroanatomy, Neuroscience Center, Goethe University, Frankfurt am Main, Germany

⦿ These authors contributed equally to this work.
* marius.schneider@esi-frankfurt.de (MS); alexander.bird@esi-frankfurt.de (ADB); cuntz@fias.uni-frankfurt.de (HC)

**Data Availability Statement:** All relevant data and code is available at https://zenodo.org/record/7863310.

## Abstract

The electrical and computational properties of neurons in our brains are determined by a rich repertoire of membrane-spanning ion channels and elaborate dendritic trees. However, the precise reason for this inherent complexity remains unknown, given that simpler models with fewer ion channels are also able to functionally reproduce the behaviour of some neurons. Here, we stochastically varied the ion channel densities of a biophysically detailed dentate gyrus granule cell model to produce a large population of putative granule cells, comparing those with all 15 original ion channels to their reduced but functional counterparts containing only 5 ion channels. Strikingly, valid parameter combinations in the full models were dramatically more frequent at ~6% vs. ~1% in the simpler model. The full models were also more stable in the face of perturbations to channel expression levels. Scaling up the numbers of ion channels artificially in the reduced models recovered these advantages confirming the key contribution of the actual number of ion channel types. We conclude that the diversity of ion channels gives a neuron greater flexibility and robustness to achieve a target excitability.

## Author summary

Over the course of billions of years, evolution has led to a wide variety of biological systems. The emergence of the more complex among these seems surprising in the light of the high demands of searching for viable solutions in a correspondingly high-dimensional parameter space. In realistic neuron models with their inherently complex ion channel composition, we find a surprisingly large number of viable solutions when selecting parameters randomly. This effect is strongly reduced in models with fewer ion channel types but is recovered when inserting additional artificial ion channels. Because concepts

**Funding:** This work was supported by BMBF (No. 01GQ1406 - Bernstein Award 2013 to HC, No. 031L0229 - HUMANEUROMOD to PJ) and by funds from the von Behring Röntgen Foundation to PJ. JT acknowledges support by the Johanna Quandt foundation and LOEWE CePTER – Center for Personalized Translational Epilepsy Research. The funders had no role in study design, data collection and analysis, decision to publish, or preparation of the manuscript.

**Competing interests:** The authors have declared that no competing interests exist.

from probability theory provide a plausible explanation for this improved distribution of valid model parameters, we propose that this effect may generalise to evolutionary selection in other complex biological systems.

## Introduction

Throughout evolution, biological cells have emerged with increasing diversity and complexity. Optimising for multiple objectives while keeping an ever larger number of cell parameters within a viable range seems a daunting task for evolutionary processes; and it remains unclear how such a multi-objective optimisation can be achieved in the corresponding high-dimensional parameter space. Here we explore the counter-intuitive hypothesis that increasing the number of mechanisms—i.e. increasing the biological complexity—potentially helps systems to evolve more quickly, easily, and efficiently towards satisfying a large number of objectives.

Neurons are a good example of complex cells, typically exhibiting a great diversity in the expression of ion channels. The channel parameters must be tuned to cooperatively generate multiple features of neuronal spiking behaviour. A palette of such spiking features has been successfully used in computational biophysical neuron models for multi-objective optimisation (MOO) using genetic algorithms [1]. Mammalian neurons contain a large variety of ion channel types in their membrane [2] producing a wide range of possible spiking mechanisms with varying temporal dynamics and excitabilities [3]. Interestingly, a number of these ion channel variants exhibit overlapping functional properties [2, 4–9]. A large body of literature has explored the reason for this high diversity and partial degeneracy [10–14]. However, it remains unclear how the diversity of ion channel types is related to the tuning of their parameters in the context of robust and flexible neuronal behaviour.

Neuronal computation relies on the morphology as well as on the diversity and distribution of ion channels in the membrane of the dendritic tree, the soma, and the axon initial segment. Even small changes in the distribution of ion channels can change the activity in neurons drastically [15]. Large differences in experimental measurements have been observed from cell to cell, day to day, and animal to animal in data from the same classes of cells [6, 10, 16–20]. The expression levels of these ion channel types can vary several-fold across neurons of a defined type [6, 10, 11, 16, 17, 19]. However, many detailed biophysical models of single cells ignore this variability in electrophysiological data and search for a fixed set of parameters that replicates an average behaviour of a particular cell type [10].

How can neurons manage to achieve a functional target activity with such a wide ion channel diversity? Using a spike-feature-based multi-objective approach, we generated large population parameter sets of dentate granule cell (GC) models with different numbers of ion channel types in order to investigate the potential advantages of ion channel diversity. We then tested to which degree the different models could compensate for pathological channel loss. Furthermore, we investigated differences in valid parameter sets, taking into account stochastic fluctuations in channel-coding gene expression. Finally, we studied the stability of the different models against ion channel alterations due to e.g. protein turnover. We found that in all cases the complete GC model with all ion channel types was more robust, stable, and had more valid parameter combinations than its reduced counterparts.

## Results

We used a recently established multi-compartmental model comprising the 15 different voltage or calcium-dependent ion channels that were described in mouse GCs [23]. The model

was specifically designed to reproduce the results not of a single experiment but of a large series of experiments and was based on raw electrophysiology traces. Its parameters were fitted to reproduce the experimental data for a number of different reconstructed (see example in Fig 1A, *Top*, from [21]) and synthetic neuronal morphologies making the model robust within the GC morphological space. Furthermore, the resulting model readily generalised to rat GCs as well as to adult born mouse GCs (i.e. GCs from adult mouse neurogenesis) after incorporating the known changes in morphology and ion channel composition. The model can therefore be considered to be robust and comprehensive. This makes it an experimentally validated tool to study the impact of complex ion channel compositions on robustness of the spiking output. To this end, we employed a population (also called "ensemble" or "database") modelling approach, which allowed us to explore the multidimensional parameter space with large populations of stochastically generated models [24–28].

## The GC model cost function

First, we developed a cost function for an automated evaluation of the validity of diverse models, which differed in their ion channel combinations and densities. Since no quantitative data exists on the particular expression of the various ion channels in individual GCs, some form of fitting procedure of channel densities was required in the construction of the GC model. The model consists of 27 conductance parameters, which precludes a comprehensive grid scan (as in e.g. [29]) for parameter fitting due to the long computing time in a 27 dimensional parameter space. The model has therefore previously been largely tuned manually with expert knowledge from GC biology. To assess the quality of any individual set of parameters more automatically, we designed a fitness function that quantified the distance to experimental spiking data (see S1 Fig for experimental data, [30]) and was inspired by approaches used previously [1, 23, see Methods]. A number of different methods have been proposed to quantify the quality of a set of parameters in relation to neuronal activity [15, 31–33]. While most studies focus on reproducing an average electrophysiological activity pattern, we wanted to focus on the distribution of valid parameter combinations in the GC model taking into account the variability present in experimental data.

We therefore used a multi-objective fitness function based on spike features, which allowed us to search for optimal trade-offs between different firing properties [1]. We extracted 9 different spiking features from raw electrophysiology traces during a 200*ms* current clamp injection with 50 and 90*pA* at the soma (Fig 1A, *Bottom*, see Methods). We then compared the values for these features between the model and the experimental data. To generate a population of GC model instances that reflected the full range of firing properties, we calculated the deviation from the experimental mean in units of experimental standard deviation (SD) [1]. In order to become a valid parameter combination in the GC model, the error value was required to be less than two SDs away from the experimental average of each feature (Eq 4, see Methods).

A manual search for parameter sets fulfilling this requirement was very time-consuming and could never be exhaustive. There are various automated parameter search methods, such as gradient descent methods, genetic algorithms, simulated annealing, and stochastic search methods, which make the search for parameters more efficient [31, 34–36]. To find a baseline model with a valid parameter combination that fulfils the cost function, we decided to use successive line minimizations in conjugate directions [37] in combination with random parameter space exploration (see Methods). The fact that some of the 9 spike features that we are optimising are correlated (*e.g.* Firing rate and ISI) did not cause a problem for the optimisation procedure as it always tries to optimise the spiking feature that shows the strongest deviation from the experimental data. This method also led to good parameter combinations within a

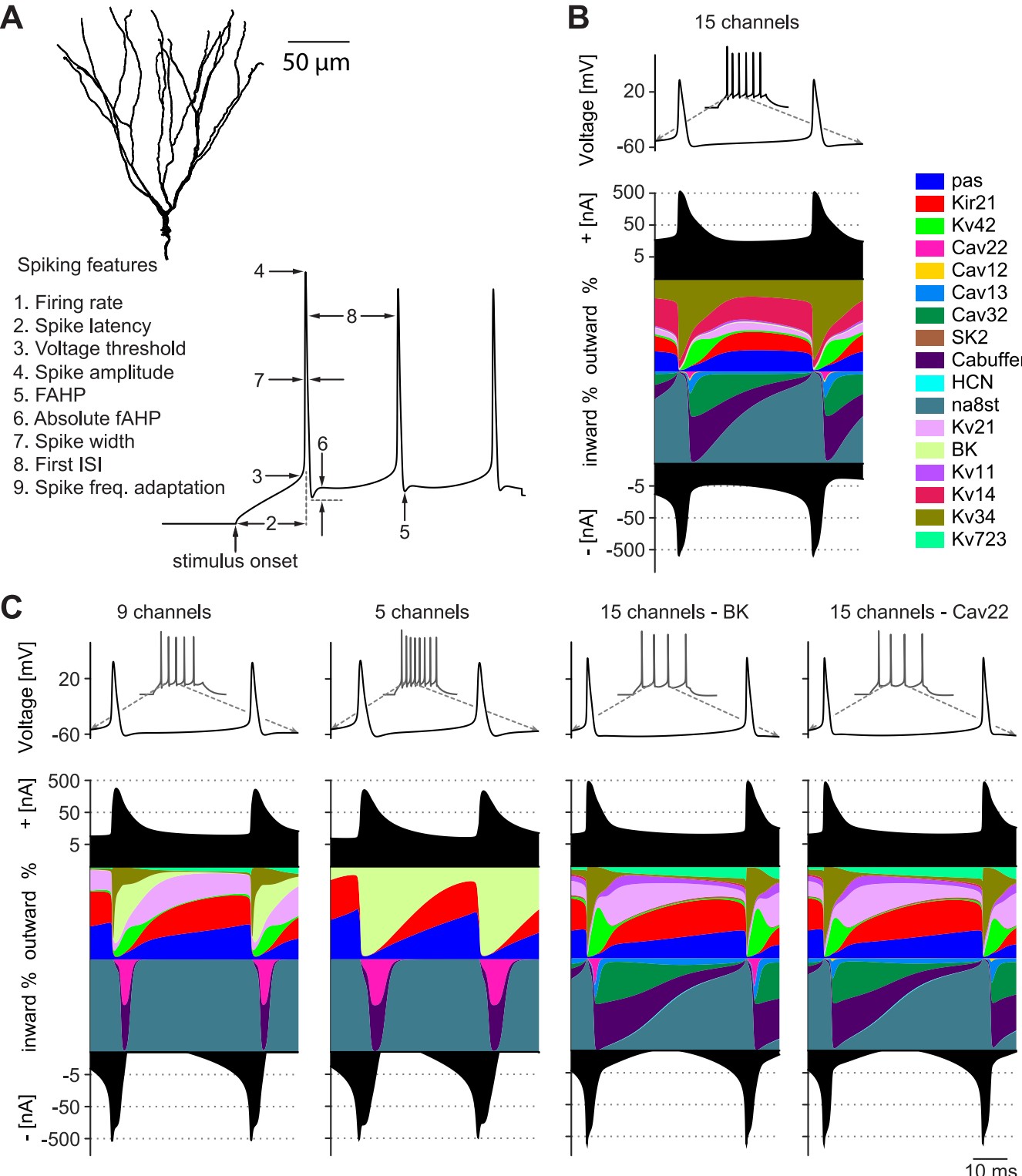

**Fig 1. Simplified as well as realistic complex ion conductance-based models capture multiple spiking features of real granule cells (GCs). A**, (*Top*) 3D-reconstructed mouse GC morphology used for our simulations [21]. (*Bottom*) Spike features used to calculate the multi-objective fitness of the GC model. **B**, Membrane potential during $200ms$ lasting current clamp of $90pA$. The coloured curves show the relative contribution of all implemented ion channels to the total inward and outward current at each time step (during the second and third spike) as a percentage of the total current. The black filled curves illustrate the total inward and outward currents on a logarithmic scale. This plot was inspired by [22]. **C**, Contribution of currents to total inward and outward current in reduced models and models that compensate for the knock out of the BK (*Left*) and Cav22 (*Right*) channel. Similar visualisation and current injection procedure as in **B**.

few iteration steps when starting from random parameter sets for which the model deviated from the experimental results. By combining random parameter exploration with successive line minimisations in conjugate directions, parameter combinations could even be found when starting from initial parameter sets for which the models produced no spikes at all (S2 Fig). Note that the optimisation algorithm was only used to generate the baseline models. The populations of valid GC models were all generated based on the baseline models by uniformly random sampling of the channel densities and subsequent counting of valid models.

## Reduction of channel diversity

Electrophysiological signatures of neurons of the same class are often unique allowing a loose classification of cell types by their electrophysiology. However, the spiking mechanisms often include multiple ion channels with overlapping functionality to achieve these specific spiking behaviours [2, 7, 9, 12, 38–40]. Thus, an important question is, how many channels are functionally necessary for a given cell type. We addressed this question in GCs whose membrane contains a large palette of voltage- and calcium-dependent conductances [23]. The compact activity together with the multitude of ion channels in the corresponding GC model (Fig 1C) suggests that a reduction of channels without losing accurate model performance might be possible. Therefore, we explored this possibility by incremental simplification of the GC model. First, we reduced the number of voltage-dependent conductances in the highly detailed multi-compartmental model of GCs by 6 channels (removing Cav12, Cav13, Cav32, Kv11, Kv14, SK, Fig 1C, *Leftmost*). This left a total of 13 parameters when expression in the different regions of the neuron are taken into account. Thereupon, we gradually reduced the number of remaining channels to a minimum of 5 ion channels (9 total parameters, leaving only the leak channels pas, as well as Kir21, Na8st, BK and Cav22); finding parameter combinations that satisfied our cost function using the search algorithm (Fig 1C, *Center left*).

To visualise the contribution of individual currents to neuronal model activity, we employed a recently developed method of plotting the time course of the relative contribution of each ionic current [22]. Overall, as expected, the electrophysiological activity of the different valid models in Fig 1C was similar (for overview, see S3 Fig). Despite the large variations in the number of ion channels, the course of the total inward and outward current flow displayed only slight changes between the three different baseline models (Fig 1B and 1C). Since GCs have a relatively simple electrophysiological repertoire (nevertheless responsible for sophisticated integration of excitatory and inhibitory information), a small number of ion channel time constants was sufficient to generate adequate firing patterns. The presence of $K^+$ and $Ca^{2+}$ channels with overlapping physiological functionality ensured that many of the channels were not crucial for the maintenance of functional activity. Only the composition of the inward and outward currents differed. In the 5−channel model, the calcium-sensitive potassium channel (BK) took over the role that 8 different $K^+$ conductances had shared in the non-reduced model (Fig 1C). BK thereby became the only remaining $K^+$ channel overall. In interaction with the $Ca^{2+}$ conductances (Cav22), the BK channel was responsible for repolarising the membrane potential following an action potential in the 5−channel model.

Recent experimental and theoretical studies demonstrated that neurons can compensate for pathological changes such as channel loss, genetic overexpression, morphological changes or increased input activity by up- and downregulation of the remaining ion channels [41–47]. This ability should be impaired in the reduced model where less redundancy exists. Indeed, we found that blocking the BK or N-type Cav22 channels in the full model was readily rescued by contributions from other channels (Fig 1C, *Right*). It is noticeable that the loss of the BK

channel was compensated by a strong upregulation of another calcium-sensitive channel (SK), as well as of voltage-dependent potassium channels (Kv 7.2/3, Kv 1.1, Kv 2.1, S4 Fig, *Left*). Neither loss of BK nor Cav22 could be compensated for in the reduced 5−channel model since it had only one active gating mechanism per ion type. Even the 9−channel model was not able to compensate for the pathological loss of Cav22 or BK. As expected, therefore, the full GC model's diversity contributed to the model's robustness with respect to the loss of specific ion channels through existing ion channel redundancies.

## Random parameter tuning as a viable approach to selecting GC model

Even though some small changes in the ion channel expression level can already lead to drastic changes in neuronal activity, several experimental studies observed that intrinsic properties of nerve cells can vary considerably across neurons of the same type [10, 16−20]. Moreover, theoretical investigations demonstrated that indistinguishable network and single neuron activity can be obtained from a large variety of model parameter settings [10, 11]. This raises the question of whether the diversity of voltage- and calcium-dependent conductances has an effect on the variability of valid parameter sets in the GC model that lead to realistic spiking activity.

In order to check this, we first generated 20,000 random model instances for each of the three baseline models by uniformly sampling the individual conductance densities within a range between 0× and 2× the value in the baseline model (all channel densities beside pas and Kir 2.1 in S1−S3 Tables were sampled). As the ohmic relations between current and voltage were consistent with experimental results in all cases (see S3(B) Fig), we did not change the densities of the leak channel or the inward-rectifying Kir21 channel, which primarily contribute to the passive properties of the neuron. The population of GC models with valid parameter combinations enabled us to calculate the Pearson's correlation coefficient $r$ for all pairs of conductance density parameters. We found weak pairwise correlations indicating low dependencies between each pair of channels and thus increasing the robustness of the model (S5 Fig). Any correlation in the parameter space restricts the space of valid solutions to the hyperspace constrained by the channel correlations. This reduces robustness as perturbations from valid solutions that don't follow the constraints given by the channel correlations end up showing non-valid electrophysiological properties. If channels are independent there is more volume in the parameter space that could be occupied by valid solutions. We made similar observations when adding correlations to the parameter space of our toy model (see Results section "Toy model points to law of large numbers"). It is likely that higher-order correlations are more prevalent in the higher-dimensional models, allowing for more different solutions that compensate for fluctuations in the expression of a single channel. The strongest pairwise correlation was observed between the expression levels of the $Na^+$ channel in the soma and in the AIS ($r = −0.95$). The sodium channel is essential for spike initiation and its presence in different regions of the GC suggests that compensatory mechanisms could simply be instantiated by maintaining a balance between the same currents in different regions, which results in a significant anticorrelation. Interestingly, the reduced models showed stronger and different pairwise correlations between the channels than the full model. This is because there are fewer mechanisms to compensate the up- or down-regulation of a specific channel, and so the mechanisms that do compensate for it must do so more strongly.

In our selection of random parameter combinations, we found valid models covering the entire sample range of the majority of parameters (Fig 2). In all cases, the most constrained parameter was the density of the 8−state $Na^+$ channel. This channel models the behaviour of all $Na^+$ conductances using a single maximum conductance parameter [21], so it is unsurprising that the neuron's behaviour is more sensitive to changes in this maximum. In addition, the

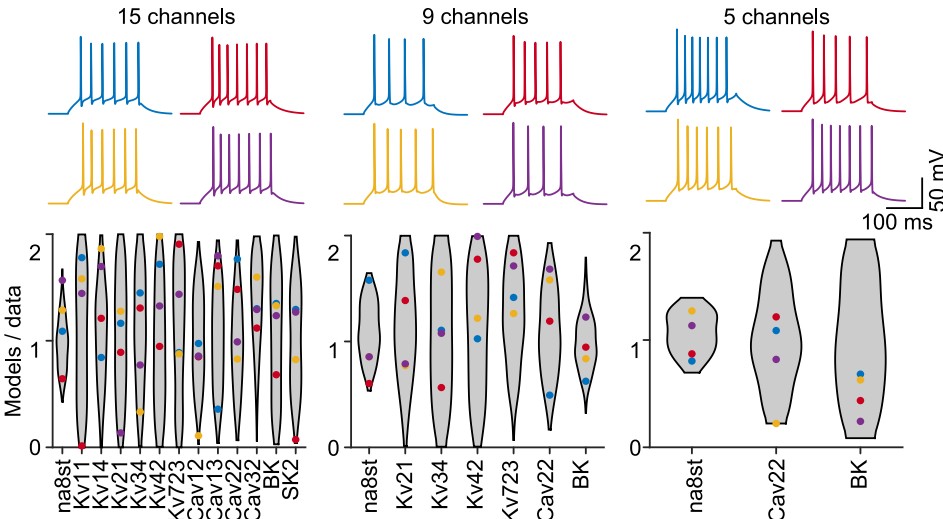

**Fig 2. Valid parameter combinations in the fully complex model are well spread.** (*Top*) Activity traces of 4 randomly picked valid parameter combinations in each of the GC models of different complexity. (*Bottom*) Coloured dots illustrate conductance densities of the four valid parameter combinations shown in top traces. The grey violin plots delimit the entire range covered by the valid parameter combinations. Conductances are weighted by the surface area of the corresponding membrane regions.

reduction of channel diversity in the 5−channel model limited the variability of the calcium-dependent potassium channel BK (Fig 2, *Right*). Surprisingly, the overall percentage of randomly selected parameter combinations that were valid increased with the number of ion channels (Fig 3A and 3B, ∼0.7% with 5 channels (for 9 total parameters), ∼3.3% with 9 channels (13 parameters), and ∼5.7% with 15 channels (27 parameters)).

The distribution of voltage- and calcium-activated channels in cell membranes is under continuous regulation [48–50]. On the one hand, the cell is subject to homeostatic regulation maintaining its electrical activity despite changes in its environment and input. On the other hand, the proteins are constantly exchanged during the lifetime of a cell. In order to investigate the stability of the valid parameter combinations in the different models in face of parameter perturbations due to e.g. protein exchange during the lifetime of a cell, we performed random walks in the parameter space. Starting from a valid parameter set that accurately reproduced the experimentally derived behaviour, we iteratively changed each parameter by random steps between −5% and + 5% of the current parameter values (counting changes in all parameters as one step). The random walk stopped as soon as the parameter combination became invalid, i.e. the cost function for the resulting model increased beyond 2 standard deviations away from experimental results. Interestingly, the average number of possible random parameter changes before model failure increased with the number of ion channels in the models (Fig 3C).

## Toy model points to law of large numbers

As shown in the previous sections, we observed an increase in valid random parameter sets when biophysical models of neurons became more complex. One possible explanation could be the fact that the more complex models included different ion channels of a similar type. Since some of these ion channels show very similar gating dynamics (see for example Cav22, Ca12 and Cav13, see Fig 1) their functional contributions may be partially redundant. A theorem from probability theory, namely the law of large numbers can play a role under such

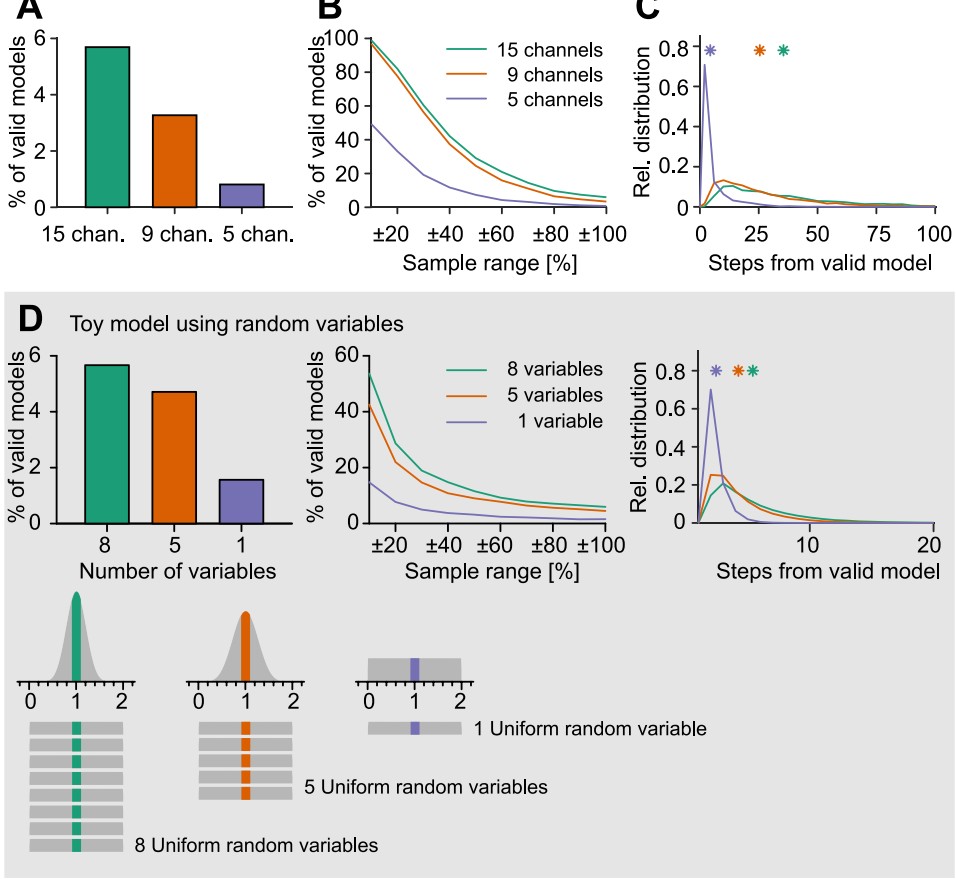

**Fig 3. Valid parameter combinations in the fully complex model are more stable as compared to reduced models.**
**A**, Percentage of valid random parameter combinations in the total population in the 2−fold range. **B**, Percentage of valid random parameter combinations in samples with different ranges around the valid reference parameter combination. Each sample contained 5,000 parameter combinations. **C**, Random walk through the parameter space starting from valid combinations in models of different complexity. Relative percentage distribution of the maximum number of random steps the respective models could undergo without losing their valid GC spiking behaviour. Bin size is 4 steps. Asterisks indicate mean number of steps the corresponding models could undergo while maintaining realistic activity. Performed for 2,000 repetitions per model. **A–C**, Colours were Green: full model, Orange: 9−channel model, Purple: 5−channel model. **D**, Reproduction of A−C with a toy model representing the model result as the average value of 1 (blue), 5 (red) and 8 (green) uniform random variables between 0 and 2. *Bottom panels*, Illustration of how the distribution of solutions becomes narrower when the number of variables is increased. This effect is explained by the law of large numbers while the Gaussian distribution results from the central limit theorem.

circumstances. The law of large numbers states that increasing the number of samples described by random variables (in our case ion channels of a similar type) will move the average over the samples closer to the expected mean value. Since in our case we sample conductances of similar ion channels, the average conductance would therefore converge towards the starting parameter set that we know is functional.

In order to illustrate this we designed a simple toy model using random variables for each parameter. Here, we represented each open parameter of the model by one random variable with a homogeneous probability of taking any number between 0 and 2 corresponding to the parameter ranges used in the neuronal model between 0× and 2× the default value (Fig 3D, *Bottommost*). To keep things simple for explanatory purposes, we set the model outcome to be the mean of the values of all separate random variables. This choice means that averaging the

parameters of a set of valid toy models would lead to another valid toy model, a situation that does not hold for complex neuronal models in general [10, see Methods section "Toy model"].

The law of large numbers predicts a decreasing variance of the mean value with an increasing number of independent random variables as illustrated in the sketch at the bottom of Fig 3D. The central limit theorem in turn predicts a Gaussian distribution for this mean over a broad range of different probability distributions for each random variable separately. In analogy to our neuronal modelling, we then constrained valid parameter combinations by a cost function allowing a maximal distance of 0.015 from the mean value, i.e. 1, averaged over all random variables. The toy model illustrates how increasing numbers of parameters can move a system relatively closer to its target if this target is its mean (Fig 3D).

Even though no strong correlations were observed in the full model's parameters (S5 Fig), they are not entirely independent, and many experimental studies have observed correlations in channel expression [19, 39, 51]. To test whether the law of large numbers can explain our observations in the case of correlated ion channel expression we added positive correlations to the parameters of our toy model. Adding positive correlations to the parameter space in the toy model did not qualitatively change the results, but stronger positive correlations decreased the proportion of valid solutions for a given number of parameters (S6(A) Fig). An important observation here is that the constraint on functionality implies negative correlations between the values of the individual random variables that make up valid points in the parameter space, despite these variables being generated independently or with positive correlations. In fact, under the toy model framework, the pairwise correlations within variables that produce valid models were almost completely independent of any correlations used to generate the overall population from which valid models are drawn (S6(B) Fig). The output correlations were instead dependent on the number of variables, with higher numbers of variables leading to weaker pairwise correlations. This result agrees with the finding of stronger pairwise correlations between channel densities in the 5-channel model compared to the full compartmental model (S5 Fig), and the mechanism, where there are fewer possible ways to compensate for changes in a given variable and so the compensations that are possible are stronger, is similar.

Correlations and relationships between input parameters do not necessarily reduce the proportion of valid solutions, however. The toy model can also be adapted to produce parameters that are more uniformly distributed in space than would be expected from an independent random process (see Methods). The 'distributed' toy model outperforms even the independent model for any multidimensional parameter space and displays a similar relationship in increasing the proportion of solutions with the number of parameters (S6 Fig). It is therefore possible that helpful relationships between input parameters could be used by an evolved system to further optimise the process of finding valid solutions to biological problems, but it is hard to distinguish evidence of this process from that of a finished optimisation that consequently imposes constraints on the relationships between its parameters as seen above.

The analogy between the toy model and the granule cell is helpful but limited since, in contrast to the channels in the GC model, all variables in our toy model are functionally the same. Moreover, the GC compartmental model applies complex nonlinear and dynamic transformations of the starting parameter space, including distinct jumps in the cost function when the model no longer produces action potentials, to reach the cost (or function) space; in the toy model the parameter and function spaces are effectively indistinguishable. However, despite its simplicity, our simple toy model was able to qualitatively reproduce all results from our GC model in Fig 3A–3C (Fig 3D). To bridge the gap between the simplest case of the toy system and the full compartmental neuron, we added attributes of a complex system to the toy model [52]. In particular, nonlinearities and random interactions between components were introduced and applied iteratively (see Methods, Eq 4). These phenomena

strengthened the effect of increasing parameter numbers on the proportion of valid solutions, with more and higher-order relationships leading to more satisfactory models (S6 Fig). The law of large numbers therefore provides a plausible explanation why a larger number of random instances, even in the more complex neuron model, would more readily linger around their target functionality.

## Additional model robustness through artificial ion channel isoforms

We have shown that the electrophysiological behaviour of GCs can be maintained despite a reduction of ion channel diversity from 15 channels to 5 channels. However, our results also suggest that this loss of ion channels goes along with a decrease in stability, a loss of compensatory opportunities, and a significant decrease in the valid model percentage within a randomised sample. From our toy model based on probability theory we postulated that it might be the mere number of ion channels that contribute to the increased robustness observed in the full model rather than the particular ion channel composition present there. In order to validate this hypothesis, we started from the reduced model and increased the number of ion channels in an artificial way to check whether we could recover the robustness present in the realistic full model.

In order to establish a quantitative relation between channel diversity and model stability in such a way, we scaled up the 5–channel model's diversity by adding more instances of the calcium (Cav22) and potassium channels (BK) remaining in that model. These artificial isoforms of the existing ion channels distinguished themselves from the original Cav22 and BK by randomised time constants (within a two-fold range of the original parameters) to allow for different dynamics through the new ion channel isoforms.

To examine the proportion of valid parameter combinations with increasing number of ion channels, we created a multitude of functional GC models with up to 20 additional ion channel isoforms (for 35 distinct channels in total). For each given number of ion channel isoforms, we randomly sampled all conductance values in a two-fold range. Thereupon we selected the three parameter combinations with the best fitness value for each number of ion channel isoforms and improved their performance by applying our search algorithm. We then followed the same procedure as in Fig 3. Using this approach, the percentage of valid parameter combinations steadily increased with the number of additional ion channel isoforms until reaching a plateau between 15 and 20 additional ion channel isoforms, for a total of 105 to 140 additional parameters (Fig 4A). To further generalise our findings in Fig 4A we have applied the same procedure to a different neuronal model type, one simulating a CA1 pyramidal neuron [53, 54, Fig 4B]. Viewed together, these results show the major contribution of ion channel diversity by demonstrating that scaling up the numbers of ion channels artificially in the reduced models leads to more frequent valid parameter combinations. This is in line with the law of large numbers.

## Discussion

We explored the complex landscape of valid parameter combinations in the parameter spaces of a detailed multi-compartmental model of dentate GCs and its simplified versions with reduced numbers of ion channels (Fig 1). We used a population modelling approach [12, 24–26] to find multiple ion channel parameter combinations for models that successfully reproduced the electrophysiological data (Fig 2 and S1 Fig). We showed that the biologically realistic GC model (full model) with many redundant ion channel types is more robust to ion channel perturbations than valid models with reduced ion channel diversity.

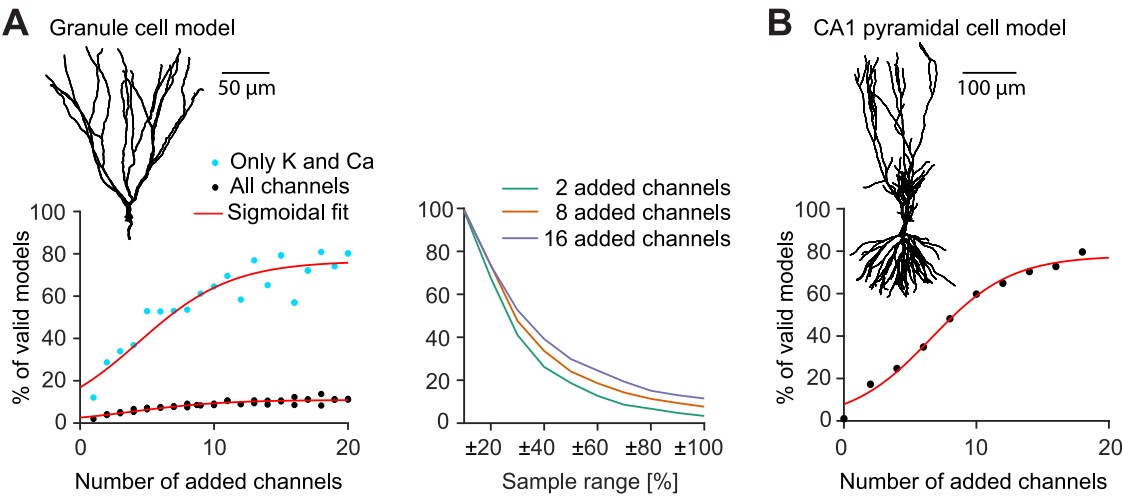

**Fig 4. Artificial expansion of ion channel diversity recovers and enhances the proportion of valid parameter combinations in the reduced 5-channel model. A**, Populations of expanded dentate GC models with 0–20 added artificial ion channel isoforms. *Left panel,* The plot shows the percentage of valid parameter combinations in a population of randomly sampled channel densities. Black dots show the populations where all ion channels (including the 8-state Markov chain modelled Na+ channel) were sampled in a 0—2 × range. Blue dots show the populations where only potassium and calcium channels were sampled in a 0—2 × range. *Right panel,* Similar plot as in Fig 3B for the black models from the *left panel*. **B**, Similar overall analysis as in **A** but for a CA1 pyramidal cell model [53].

## Robustness through ion channel degeneracy in complex GC models

Most neurons contain more than a dozen different ion channels. While early computational models implemented considerably fewer channels than known in biology, more and more models exist that contain a realistic number of mechanisms e.g. [23, 55]. Although the different potassium channels in mammalian cortical neurons differ genetically, some are remarkably similar in their functional contribution to the electrophysiological activity of neurons [2, 40]. This functional similarity is often referred to as degeneracy [9] and is not a phenomenon restricted to neurobiology [56, 57]. Depending on the computations a neuron should implement, its dynamics only need to cover certain relevant time scales, e.g. in the form of different time constants of its gating variables [58]. Since five channels were sufficient to support realistic voltage dynamics at relevant time scales, we were able to reduce the original variety of ion channels without observing a significant loss in the performance of the model. In our study, GCs with their compact electrophysiological repertoire did not require a large variety of ion channels to reproduce their characteristic activity patterns. To replicate the 9 experimentally derived spiking properties, the models required only one active channel of each of the different subgroups of ion channels (one Na$^+$-, one K$^+$- and one Ca$^{2+}$-channel, as well as the leak channels; Fig 1C).

Experimental as well as theoretical studies from the last decades revealed that pharmacological manipulations like the blockage or upregulation of intrinsic or synaptic mechanisms, resulting in a pathological cellular activity on a short timescale, can be compensated by up- and downregulation of the remaining conductances on a long timescale [17, 18, 39, 40, 43, 44, 47, 59]. Interestingly, not all manipulations can be compensated by mechanisms of homeostatic regulation [60, 61], indicating differences in the capability of homeostatic compensation between ion channels as well as types of neurons. As opposed to other studies using biophysically realistic mechanisms of homeostatic intrinsic plasticity based on calcium signals (13, 39, 62–65; see also 61), we decided to use a successive line minimisation in conjugated directions

approach to investigate the large and complex parameter space of possible intrinsic compensations. A homeostatic mechanism based on a single feedback signal [13, 39] that has been suggested to play a role in model robustness was not compatible with our model in our hands since it consistently led to decreased ion channel degeneracy. This is in agreement with a recent study [61] that provided new insights into the complex relationship between ion channel diversity and homeostatic co-regulation of ion channel densities. The study by [61] suggested the necessity of more than one master feedback regulator (i.e. more regulators than just global calcium) for homeostatic feedback loops, which must co-tune numerous degenerate and pleiotropic ion channels to achieve multiple regulated functions or objectives (cf. 28, 66). Viewed together, we believe that diversity and (multi-signal) feedback can act as independent mechanisms to ensure viable and robust solutions to the multi-objective optimisation problems of neurons.

We demonstrated that the full GC model was capable of compensating the loss of any potassium and calcium channels by up- and downregulation of the remaining ion channels (Fig 1C). In contrast, the different reduced models relied on the presence of certain indispensable ion channels, without which they could not capture the main electrophysiological characteristics of GCs. S4 Fig shows that there can be as much as a 20−fold variability in the density of voltage-dependent ion channels. Experimental studies have observed variations of a similar order of magnitude as a result of compensatory mechanisms [17]. The ability of these models to compensate for losses of ion channels can be attributed to the overlapping or degenerate physiological function of the present potassium and calcium channels [67].

The reduction of the diversity of gating mechanisms goes along with a loss of space to manoeuvre in the process of achieving functional target activity [13, 40]. In case of a loss of the BK channel, several potassium channels (see S4 Fig) were upregulated, and thus maintained the functional behaviour of the cell. In line with the concept of degeneracy [1, 42], the overlapping functionality of different channels enabled the neuron, depending on the given conditions, to achieve a target spiking behaviour in a number of different ways.

In addition, we tested the stability of the differently reduced models against random parameter perturbations, in order to simulate putative protein exchange during the lifetime of a cell. The ongoing protein replacement is one of the reasons for the continuous regulation of voltage- and calcium-dependent channels in cell membranes [39, 48, 49]. Although no homeostatic tuning mechanism with dynamic feedback was implemented, valid parameter combinations in the complete model were able to endure far more random parameter perturbations while maintaining realistic activity than the ones in the reduced models (Fig 3C). This is in agreement with experimental studies, which have shown that, although homeostatic tuning rules can compensate for many perturbations and knock-outs of ion channels, not all channel deletions and perturbations can be compensated for [60]. A challenge for future experimental work will be to uncover the long-term effects of ion channel knock-outs in GCs in order to find out whether our theoretical results of the outstanding robustness of GCs against channel deletions can be observed in biology.

## Random parameter selection as a viable fitting strategy for neurons

Like many biological processes, gene expression is a largely stochastic process resulting in considerable heterogeneity of mRNA and protein levels [48, 49, 68]. This noise in gene expression is one reason for the cell-to-cell variability. However, noise in gene expression could be harmful for achieving valid parameter sets of ion channel expression during developmental maturation or during pathological perturbations. Neurons are thought to target certain desired subspaces in the output space (i.e. function space, e.g. defined by firing properties)

corresponding to valid subspaces in the high-dimensional parameter space of ion channel properties [28]. Our simulations show that the subspace around these target values in parameter space tends to be more densely filled with valid model parameters than non-valid parameters (Fig 3B), particularly in higher dimensions. Accordingly, despite fluctuations, high-dimensional models are more likely to end up in functional subspaces. Even without the implementation of homeostatic regulation processes, the chance of obtaining a valid ion channel expression level is relatively high. This implies that the degeneracy between ion channel types and isoforms supports robust excitability profiles in neurons despite random fluctuations in the expression of ion channels. Our computational analysis indicates that a complex high-dimensional parameter space supports the stability of neuronal excitability against perturbations that would push neurons into non-functional subspaces. The reason for this is that the volume of valid solutions in the parameter space around a valid parameter set is larger in higher-dimensional parameter spaces than in lower-dimensional ones. This in turn increases the likelihood of neurons returning into functional subspaces by random ion channel parameter adjustment.

An interesting extension would be to compare the efficiency of activity-dependent regulation [39, 61, 65] implemented with single or multiple homeostatic error signals [61], with the multi-objective optimisation [1, 28, 66, 69] that arises naturally from stochastically exploring high-dimensional parameter spaces.

Due to the diversity of electrophysiological mechanisms, the cell is able to generate valid electrophysiological activity by random selection of parameters with a high chance of success despite stochastic fluctuations in the expression of channel-coding genes. We showed that there was a clear relation between the number of intrinsic mechanisms and the chance to obtain a valid set of parameters from a random sample around a valid point in parameter space that produces functional activity in output space (Figs 3A, 3B and 4). Furthermore, we showed that many other parameter combinations existed around such a functional point in the parameter space that fulfilled our criteria for functional activity. While in a random $0 - 2$-fold sample of the initial model, about $\sim 5.7\%$ of the parameter combinations showed valid GC activity, this proportion decreased steadily to $\sim 0.7\%$ with a reduction of the model (Fig 3A). In the closer surrounding of the baseline models this difference was even more obvious. While in the unreduced model in the close neighbourhood of $\pm 20\%$ of the initial parameter sets over 80% of the models showed characteristic GC activity, in the heavily reduced model it was only about 30% (Fig 3B).

Similar to [15] and [7] we showed that near each functional point in the parameter space many other parameter sets exist whose activity matches the activity of the original parameter set (S7–S9 Figs). Instead of talking about parameter sets, one might rather speak about subspaces that show functional behaviour. These subspaces can have different densities of parameter sets showing characteristic electrophysiological activity. This depends to a great extent on the diversity of the channels (Figs 3A, 3B and 4A, *Left panel*). Furthermore, different valid subspaces with the same diversity differ in their density of valid solutions located in this subspace. In order to be as robust as possible against perturbations and to simplify the process of parameter fitting, it seems reasonable for a neuron to target as densely populated a subspace as possible.

## Ion channel correlations and random expression

When analysing the conductance values of the different types of ion channels in the valid models, we observed that some pairs of ion channels shared significant correlations (S5 Fig, *Red squares*). This is in line with experimental studies of cell-to-cell variations in ion channels showing that some ion channels are co-expressed and might be co-regulated [19, 20, 51, 70–

73]. Future large-scale analysis of channel expression in real populations of GCs might validate the diversity of and correlations between expression levels in our population models.

In our simulations, the ion channel correlations arose from constraints on the resultant functionality because our model-generating strategy sampled the ion conductance levels independently. Although our population modelling was inspired by random noise in gene expression, it does not imply that random noise is the only or predominant source of cell-to-cell variability in ion channel expression. Since the above mentioned experimental studies found ion channel co-expression, it is likely that a great amount of the cell-to-cell variability in ion channel expression is due to transcription regulatory mechanisms, and only to some extent to the unreliable and noisy nature of gene expression mechanisms. Indeed, our toy model hinted at ways in which relationships between levels of expression could further enhance the proportion of functional neurons. Widespread ion channel co-variations might also potentially arise from homeostatic feedback mechanisms (13, 39, 61, 65; see above). These observations and models do not undermine our modelling strategy, but complement and extend our assumption that some of the variability in ion channel expression is due to intrinsic noise in the expression machinery.

## Probabilistic toy model and law of large numbers

We have put forward the law of large numbers as a possible explanation for our observations in the GC model. As a consequence of the law of large numbers, a model containing more ion channels tends to exhibit a behaviour that is closer to its expected target behaviour (Fig 3). This effect was not qualitatively affected by either positive or negative correlations between components, or by the imposition of interactions and nonlinearities that would be expected in a complex system such as a biological neuron [52]. Accordingly, we were able to recover the amount of robustness observed in our full compartmental model when adding artificial ion channel isoforms to the reduced model (Fig 4). This is a strong indicator that indeed the number of ion channels and not their specific composition or spanning of function space leads to the effect that we observed.

However, this interpretation is not mutually exclusive to the complementary insight from biophysical modelling that the 15-channel model is more robust than the 5-channel one due to the increasing timescale and voltage coverage with the increasing number of ion channels (due to the partial, but not complete, redundancy between similar ion channels). The abstract toy model does not account for these two (time and voltage-related) mechanistic aspects but offers an intuition for the impact of the number of ion channel instances and their stochastic variation. The increase in the number of random variables or their interactions in the toy model is analogous to the increase in the number of random instances of different ion channels. The main biological insight from the toy model is that if a neuron samples conductances of similar ion channels around a valid point in parameter space, with the increasing number of channels the average conductance will converge towards the valid parameter set that produces functional behaviour despite the complexity of the interactions between channels or their correlations.

In summary, both biophysical and toy models indicate that the large number of ion channel subtypes and isoforms expressed by a neuronal type supports the tuning and robustness of the electrophysiological phenotype.

## Conclusions and outlook

Overall, our results suggest that the diversity of ion channels allows for increased robustness and higher flexibility in finding a functionally valid solution in the complex parameter space of

a neuron's conductances. It will be interesting to investigate whether our findings here translate to other biologically complex systems, in which case they will most likely affect our general understanding of how evolution deals with complex dynamics [74].

## Materials and methods

All simulations were performed in *Matlab 2017b* (Mathworks, Natick, MA, USA). Single neuron simulations were performed using *T2N* [23, www.treestoolbox.org/T2N], a Matlab interface between the open source package *TREES toolbox* [75, 76, www.treestoolbox.org] and the *NEURON* simulation environment [77, www.neuron.yale.edu]. Predefined functions from *TREES toolbox*, *T2N* as well as additional custom *Matlab* code were used to generate and analyse the models.

### The granule cell (GC) model

The GC model used in this study has been fully described in [23]. Briefly, the model was designed to reproduce passive and active GC properties as determined by voltage and current clamp experiments, dendritic patch recordings of bAPs, and intracellular calcium imaging. In order to reduce the number of parameters and to speed up simulations we simplified the morphology by deleting the artificially added axon. The loss of the axon was compensated by slight changes of the maximum conductances in the axon initial segment (AIS). Since the HCN channel in its original form had no influence on control GC activity, we did not take it into account. The compartment-specific distributions of ion channels are shown in S1 Table. Detailed descriptions of the individual ion channels can be found in [23]. We used a realistic three-dimensional granule cell morphology from [21].

### Stimulation protocols and cost function

Instead of using a single optimal error function, we decided to adopt a strategy that allows to take into account several potentially important properties of GC activity. To get a first impression of the "goodness of a model", we compared the experimental [30] and the model spiking-properties following a 200*ms* current injection of 50 or 90*pA*. The stimulation protocol was as follows: 50*ms* prerun without stimulation, followed by 200*ms* somatic current injection of 50 or 90*pA* followed by a 50*ms* long period without current injection.

We extracted the following 9 spiking properties (Fig 1A) from the raw traces of current injections with 50 and 90*pA*:

1. Numbers of spikes fired within 200*ms* under current clamp.

2. Latency of first spike after stimulus onset in *ms*.

3. The voltage threshold was defined as the voltage at which the rate of change of membrane potential exceeded $15\frac{mV}{ms}$.

4. Average amplitude of spikes.

5. The fast after hyperpolarisation (fAHP) amplitude was calculated as the voltage difference between the spiking threshold and the minimum potential within 5*ms* after a spike.

6. Absolute value of fast after hyperpolarisation (fAHP) amplitude.

7. The action potential width was measured at half the height of the spike amplitude.

8. Interspike interval (ISI) in *ms* between the first and second spike during current clamp.

9. The adaptation index $AI$ was calculated in the following manner: $AI = 1 - \frac{ISI_1}{ISI_{end}}$, where $ISI_1$ is the first and $ISI_{end}$ the last ISI.

The spiking features for any given parameter combination in the model were then compared with the same experimentally derived spiking features [30] and expressed in units of standard deviation. This approach allowed us to take into account the intrinsic variability of each feature separately. The overall fitness $F_i$ of spike feature $i$ was defined as:

$$F_i = \frac{|SF_i - \overline{SF}_{i,exp}|}{SD_{i,exp}} \tag{1}$$

where $\overline{SF}_{i,exp}$ refers to the average value of the spike feature $i$ and $SD_{i,exp}$ to the standard deviation of the spike feature $i$ across all recorded GCs. The value of the spike feature of the corresponding model for a given parameter combination was $SF_i$. For a parameter combination to be accepted as a valid combination, it was required to fulfil the following condition:

$$P = max\left(\frac{|SF_i - \overline{SF}_{i,exp}|}{SD_{i,exp}}\right) < 2, \; for \; i = 1, 2, \ldots, 9 \tag{2}$$

The value of the Pareto efficiency $P$ corresponded to the fitness $F_i$ of the spiking feature $SF_i$ that deviated most from the experimental average.

## The search algorithm

To search for parameter sets that match our criteria for valid GC activity we combined random sampling with successive line minimizations in conjugate directions [37]. Starting from a random or given point in the parameter space, we evaluate the change of the cost function for each dimension with two sample points to smooth the slopes. The algorithm evaluates the fitness function in each dimension and moves in the direction of the steepest descent with respect to the cost function. The sample points where calculated in steps of ±5% of the corresponding parameter value. This procedure was then repeated until the method converged to a local minimum of the corresponding Pareto efficiency $P$ (Eq 2). The successive line minimisation was done in conjugated directions, so that the successive minimisations were as independent as possible. Theoretically, this ensured that the parameter search found a local minimum of the target function $P$. For some initial parameter combinations, large areas of the parameter space were completely flat (i.e. the gradient was zero). This was especially the case when the initial models showed no spiking activity (S2 Fig). In this case, we increased the size of the iteration steps consecutively by ±5%. If still (after increasing the step size to ±50%) no gradients other than zero were found or the local minima did not fulfil the criteria of functional GC excitability, we randomised the parameters in the next step in an iteratively increasing range (from ±10% of the corresponding parameter values in steps of ±10% up to ±50%). This approach also helps us with the fact that some spiking features that we optimise have discrete integer values (whose gradients are not smooth and finite). The search algorithm was used to find the parameter settings of the reduced models. Starting from the full model (Fig 1C and S1–S3 Tables), we gradually reduced the number of ion channels, starting with the channels that influenced the cost function the least.

## Diversity expansion

In order to generate models with controllable amounts of ion channels we used the reduced 5-channel model as a basis. We then produced multiple instances of each of the remaining

potassium (BK) and calcium (Cav22) channels. Each artificial channel form obtained in this way was assigned a randomised time constant, which was uniformly sampled between $0 \times$ and $2 \times$ the value in the original GC model, in order to obtain altered dynamics. Furthermore, we randomised the conductances and applied the search algorithm to reproduce characteristic GC activity to derive all base models with different complexities in Fig 4.

## Toy model

We created a toy model to test whether the law of large numbers is a plausible explanation for the phenomena we observed in the GC model. In order to mimic the distribution of functionally overlapping ion channel expressions in a population of GC models around a genetically targeted functional set point we used randomly uniformly sampled variables between zero and two (Fig 3D). A valid toy model is defined as having a smaller average deviation from the mean (targeted value) than 0.015. By decreasing the sample range around the mean in steps of 0.1 down to a sample range between 0.9 and 1.1 we changed the intensity of fluctuations around the target point (Fig 3D).

Golowasch et al., 2002 [10] showed that taking average parameter values over a set of valid conductance-based neuronal models would typically produce invalid 'average' models, as the valid models were constrained to a concave region of parameter space that did not contain the mean. There is no direct discrepancy between this paper and our toy model, where averaging is successful, as the toy model is constructed in such a way that the 'average' model would lie within (and at the very centre of) the space of valid models. Averaging over valid solutions for the full and reduced models (Fig 2) would, similarly to what Golowasch et al., 2002 [10] found, not necessarily lead to valid solutions.

To expand the toy model to account for possible intrinsic correlations in the expression of ion channels (S6 Fig), we used a Gaussian copula to impose a correlation structure on the random variables with uniform marginals and specified pairwise correlations. For a desired positive pairwise correlation $\rho$ in a system of $n$ variables we generated an $n \times n$ correlation matrix $\mathbf{R}$ with elements $\mathbf{R}_{i,j} = \rho$ if $i \neq j$ and $\mathbf{R}_{i,i} = 1$. If a random variable $\mathbf{u} = (u_1, u_2, \ldots, u_n)$ and each $u_i$ is independently uniformly distributed in the range $[0, 1]$, then the correlated random variable $\mathbf{v}$ with uniform marginals on $[0, 1]$ is given by

$$\mathbf{v} = \Phi_{\mathbf{R}}(\Phi^{-1}(u_1), \Phi^{-1}(u_2), \ldots, \Phi^{-1}(u_n)) \tag{3}$$

where $\Phi_{\mathbf{R}}$ is the cumulative distribution function of a multivariate Gaussian distribution in $n$-dimensions with mean 0 and covariance matrix $\mathbf{R}$ and $\Phi^{-1}$ is the inverse cumulative distribution function of a standard univariate Gaussian. Multiplying $\mathbf{v}$ by 2 maps it back to the same space as the uncorrelated toy model.

As it is not possible to specify an arbitrary pairwise negative correlation for elements of a high-dimensional vector, a different approach was necessary to generate 'distributed' systems of variables. The algorithm used increases the mean distance between each variable and its nearest neighbours. To begin, a random variable $x_1$ is chosen uniformly on the interval $[0, 2]$. If $x_1 \geq 2 - x_1$, then the next variable $x_2$ is chosen uniformly randomly from $[0, x_1)$, otherwise it is chosen uniformly randomly from $(x_1, 2]$. The next random variable is chosen uniformly randomly from the largest interval between the (ordered) existing random variables and so on until $n$ variables have been chosen. The sum of a vector generated in this manner is typically closer to $n$ than for an uncorrelated vector of the same size (S6(C) Fig, dashed line).

To simply model interactions between components of the model and nonlinearities, as are found in a complex system such as a biological cell, vectors of random variables are multiplied by an interaction matrix and passed through a sigmoidal activation function. Concretely, let $\mathbf{u}$

be an $n \times 1$ uniform random vector with elements drawn from the interval $[0, 2]$, $\mathbf{M}$ and $\hat{\mathbf{M}}$ be $n \times n$ random matrices each with elements drawn uniformly from the interval $[-1, 1]$ and normalised columnwise to make $\mathbf{M}$ and $\hat{\mathbf{M}}$ left-stochastic, and $S(\mathbf{x}) = -1 + \frac{2}{1+\exp^{-\mathbf{x}}}$ be an element-wise sigmoid function on a vector $\mathbf{x}$. Then an $0^{th}$- order vector $\mathbf{v_0}$, a $1^{st}$- order vector $\mathbf{v_1}$, and a $2^{nd}$- order vector $\mathbf{v_2}$ are given respectively by

$$\mathbf{v_0} = S(\mathbf{u}) \quad , \quad \mathbf{v_1} = S(\mathbf{M} . S(\mathbf{u})) \quad , \quad \mathbf{v_2} = S\left(\hat{\mathbf{M}} . S(\mathbf{M} . S(\mathbf{u}))\right) \tag{4}$$

Increasing the order of the interactions and the number of nonlinearities can lead to a higher proportion of valid models (S6(D) Fig).

## Hyperplanes

To learn more about the relationship of the set of valid models, we created linear combinations of our best solutions. This method was adopted from [15] and allowed us to better estimate whether the solutions lie on a common low-dimensional manifold within the high-dimensional parameter space of the GC model variants (S7–S9 Figs). As a first step, we created linear combinations out of weighted sums of a pair of solutions. We weighted the parameters of the respective model between 0.1 and 0.9 with a step size of 0.1. The weighting of the second solution was chosen such that the sum of the weights was equal to 1. As soon as the Pareto efficiency of all evaluated linear combinations fulfilled the criteria for characteristic GC spiking, we assumed that the respective models were connected. In the next step, we created linear combinations of three different valid solutions to visualise the hyperplanes in two dimensions. We used several triplets of valid parameter sets and weighted two of them with values between −1.5 and 2.5 using a step size of 0.04. The corresponding grid of combinations was visualised in a two-dimensional plot. The weighting of the third selected parameter set was chosen in a way that the sum of all weights was equal to 1. The hyperplanes consisted of several thousand points, whereby the parameter sets with negative values were removed. As a result, each hyperplane had different boundaries and thus a different size. Finally, for each of these points we ran simulations and calculated their Pareto efficiency. The Pareto efficiency of the models without spiking behaviour was set to 6, which explains the abrupt change of colour on the right side of S7 Fig. The colour selection of the plots allowed a clear distinction between the valid (green) and the nonvalid (blue) models.

## Supporting information

**S1 Fig. Electrophysiological properties of mouse GCs.** Experimental data from [30]. **A**, Voltage traces of eight different GCs during 200$ms$ current clamp injection of 90$pA$. **B**, Frequency of action potentials elicited by 200$ms$ lasting current injections (mean and standard deviation from raw traces, experimental standard deviation is shown as grey patches). **C**, Current-voltage (I–V) relationships (mean and standard deviation from raw traces, experimental standard deviation is shown as grey patches). **D**, Phase plots of the first action potential during 90$pA$ current clamp. Modified from Fig 2 in [23].
(PDF)

**S2 Fig. Gradient descent using multi-objective optimisation. A**, Temporal evolution of Pareto optimality (*top*, see Eq 4) using the gradient descent method. Solutions are considered valid once their Pareto optimality drops below 2 (dashed line). Initial parameter combinations are random non-valid parameter combinations within a range between 0× and 2× the value in the reference parameter set. (*bottom*) Voltage traces of the model with initial parameter

combinations (grey) and optimised parameters (green). **B**, Same as in **A**, but all initial parameter combinations were in a similar order of magnitude of Pareto optimality with corresponding models that did not even produce spikes.
(PDF)

**S3 Fig. Comparison of the different GC models in Fig 1C. A–D**, Similar panels as in S1 Fig for the different models and respective parameter combinations as in Fig 2A.
(PDF)

**S4 Fig. Valid parameter combinations in models that compensate for the knock-out of the BK (Left) and Cav22 (Right) channel.** Valid parameter combinations in the fully complex model are well spread and more stable as compared to reduced models. Activity traces of 4 randomly picked valid parameter combinations in models successfully compensating the corresponding knock-out (Top). Coloured dots illustrate conductance densities of the four valid parameter combinations shown in top traces (Bottom). Violin plots show the probability distribution of valid parameter combinations. Conductances are weighted by the surface area of the corresponding membrane regions.
(PDF)

**S5 Fig. Correlations between pairs of channel conductances in the different populations.** Significant correlations are highlighted by red boxes (p-value <0.01). Pairwise correlations in population of **A**, 15−channel models, **B**, 9−channel models, **C**, 5−channel models.
(PDF)

**S6 Fig. Correlations and complexity in the toy model. A**, Effects of pairwise correlations on the proportion of valid models for different numbers of variables. The 1 variable model is not plotted as it is not affected by correlations. All models converge to the same point as their elements become perfectly correlated and the effective number of dimensions is reduced to 1. **B**, Observed output correlations in valid models as a function of the pairwise correlation used to generate the population from which valid models are drawn. For almost all input correlations the observed correlation depends only on the number of variables. **C**, Distributing parameters more evenly in space (dashed line) led to even more solutions than in the independent model (solid line) and the positively correlated model (dotted line). **D**, Adding hierarchical interactions and nonlinearities improved the validity of the models (Eq 4). $\mathbf{v_0}$ (solid), $\mathbf{v_1}$ (dashed), and $\mathbf{v_2}$ (dotted).
(PNG)

**S7 Fig. 2D illustrations of hyperplanes in the parameter space.** Hyperplane analysis inspired by [15] for the 15−channel model. **A**, The hyperplane of **B** is shown in red as projection onto $g_{Na8st,AIS}$ vs. $g_{SK2,AIS}$ plane. 25 randomly chosen valid parameter combinations are represented by dots. The blue hyperplane is parallel to the red and is defined by the addition of 10% of the SD of all solutions (in every dimension). **B**, Hyperplane defined by the three individuals on the red line in **A**. The Fitness of all points is colour scaled. The three original individuals are highlighted as red dots. **C**, The red dots mark the places parallel to the 3 originally selected individuals.
(PDF)

**S8 Fig. 2D illustrations of hyperplanes in the parameter space.** Hyperplane analysis inspired by [15] for the 9−channel model. **A**, The hyperplane of **B** is shown in red as projection onto $g_{Kv34,AIS}$ vs. $g_{Cav22,AIS}$ plane. 25 randomly chosen valid parameter combinations are represented by dots. The blue hyperplane is parallel to the red and is defined by the addition of 10% of the SD of all solutions (in every dimension). **B**, Hyperplane defined by the three individuals on the

red line in **A**. The Fitness of all points is colour scaled. The three original individuals are highlighted as red dots. **C**, The red dots mark the places parallel to the 3 originally selected individuals.
(PDF)

**S9 Fig. 2D illustrations of hyperplanes in the parameter space.** Hyperplane analysis inspired by [15] for the 5−channel model. **A**, The hyperplane of **B** is shown in red as projection onto $g_{na8st,AIS}$ vs. $g_{Cav22,AIS}$ plane. 25 randomly chosen valid parameter combinations are represented by dots. The blue hyperplane is parallel to the red and is defined by the addition of 10% of the SD of all solutions (in every dimension). **B**, Hyperplane defined by the three individuals on the red line in **A**. The Fitness of all points is colour scaled. The three original individuals are highlighted as red dots. **C**, The red dots mark the places parallel to the 3 originally selected individuals.
(PDF)

**S1 Table. Summary of ion channel densities and models implemented in the 15-channel model.** Ion channels and their expression profiles in the corresponding morphological compartments. Conductance densities are given in units of $^{mS}/_{cm^2}$.
(PDF)

**S2 Table. Summary of ion channel densities and models implemented in the 9-channel model.** Ion channels and their expression profiles in the corresponding morphological compartments. Conductance densities are given in units of $^{mS}/_{cm^2}$.
(PDF)

**S3 Table. Summary of ion channel densities and models implemented in the 5-channel model.** Ion channels and their expression profiles in the corresponding morphological compartments. Conductance densities are given in units of $^{mS}/_{cm^2}$.
(PDF)

## Author Contributions

**Conceptualization:** Marius Schneider, Alexander D. Bird, Albert Gidon, Jochen Triesch, Peter Jedlicka, Hermann Cuntz.

**Data curation:** Marius Schneider, Alexander D. Bird.

**Formal analysis:** Marius Schneider, Alexander D. Bird.

**Funding acquisition:** Peter Jedlicka, Hermann Cuntz.

**Investigation:** Marius Schneider, Alexander D. Bird.

**Methodology:** Marius Schneider, Alexander D. Bird.

**Project administration:** Peter Jedlicka, Hermann Cuntz.

**Resources:** Peter Jedlicka, Hermann Cuntz.

**Software:** Marius Schneider, Alexander D. Bird.

**Supervision:** Jochen Triesch, Peter Jedlicka, Hermann Cuntz.

**Validation:** Marius Schneider, Alexander D. Bird.

**Visualization:** Marius Schneider, Alexander D. Bird.

**Writing – original draft:** Marius Schneider, Alexander D. Bird, Jochen Triesch, Peter Jedlicka, Hermann Cuntz.

**Writing – review & editing:** Marius Schneider, Alexander D. Bird, Albert Gidon, Jochen Triesch, Peter Jedlicka, Hermann Cuntz.

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
