## [Decision Letter · Decision Letter 0]

24 Mar 2023

Dear Mr Schneider,

Thank you very much for submitting your manuscript "Biological complexity facilitates tuning of the neuronal parameter space" for consideration at PLOS Computational Biology. As with all papers reviewed by the journal, your manuscript was reviewed by members of the editorial board and by several independent reviewers. The reviewers appreciated the attention to an important topic. Based on the reviews, we are likely to accept this manuscript for publication, providing that you modify the manuscript according to the review recommendations.

Sincerely,

Joanna Jędrzejewska-Szmek, Ph.D.

Academic Editor

PLOS Computational Biology

Daniele Marinazzo

Section Editor

PLOS Computational Biology

Reviewer's Responses to Questions

**Comments to the Authors:**

Reviewer #1: Overview: This is an important paper that addresses a fundamental set of questions relevant to the expression of ion channels in neurons. The direct comparisons of the 15 channel conductance model to the 5 conductance model is quite lovely. One of the results I was initially surprised by, but makes sense after thinking, is that the reduced models showed stronger and different pairwise correlations than the full model.

Specific Comments:

1.Should Swensen (2005) be Swensen and Bean (2005)?

2.Some of the 9 features that were used might be correlated? So for example, firing rate and ISI? Is this important? Does it matter?

3.In my mind, the business with the law of large numbers is just a distraction from the beauty of the results. But I understand that the authors might disagree strongly, and that is fine. I just am asking them to think through this once again.

4.The first paragraph of the Discussion is just a recap of the results. I understand many people do this, but to me this is already in the Abstract and Results, and doesn’t need to be repeated. Instead the first paragraph of the Discussion should set the conceptual stage and framework for the work. This is especially true because the Discussion is already long.Moreover, other parts of the Discussion can probably be shortened a bit, and would enhance the force of the new findings rather than just reiterating them?

Reviewer #2: The paper of Schneider et al. addresses an important question in computational neuroscience, namely what is the best level of model complexity when modeling neurons. In particular, they study whether a 15-channel model of a hippocampal granule cell , with its large number of a priori unknown parameters, nevertheless is preferable to a simple 5-channel model with fewer unknown parameters. The claim is that it is, at least when comparing the fraction of acceptable ("valid") model parameter combinations found from randomly perturbing parameters around a "valid" starting solution.

I find the problem and results to be both important and timely, and I find the manuscript highly suited for PLoS Computaional Biology. However, I have a "major" question and some "minor commment/questions" that I would like to see addressed by the authors.

Major:

- On p. 8 it reads: "Since we were starting from a valid parameter combination, we decided to use a gradient descent algorithm (Press et al., 2007) in combination with random parameter space exploration (see Methods)." Using gradient descent seems a bit unsuited to answer how many good solutions there are in the parameter space in general. Instead of seeing how many solutions there are in general, doesn't it rather check how easy it is to find solutions? Couldn't the observation be explained by the gradient descent working better in high-dimensional system since the chances of being stuck in a local minimum is lower (since the chances of being stuck at local error-minimum with a valley in all directions is lower the more dimensions there are). As I understand it, this has been used as an argument for why one often finds a good minimum when fitting deep networks, see Sejnowski et al, PNAS, 2020 (https://www.pnas.org/doi/10.1073/pnas.1907373117). By this reasoning it seems that sampling parameter space in a more random way around the "valid solution" is necessary to diambiguate the current interpretation in the paper from the alternative possibility that it is easier to find good solutions using gradient descent in high-dimensional neural models (which also would be an interesting result). I suggest that the authors pursue this.

Minor points:

- I think the model for completeness should be included in an Appendix or as Supplementary material

- It should be clearly stated somewhere what paramters are allowed to vary (only channel densities I presume?)

- p. 8: I do not see how the referenced figure S1 helps explain the fitness function that was used.

- On p. 9 it says ".. a small number of membrane time constants ...". This is confusing: Isn't the membrane time constant defined as the product of the leak membrane resitance and the membrane capactitance?

- p. 11: I am not sure I follow why channel independence implies robustness? Maybe expand more on this idea.

- p. 21: Difficult to understand following sentence: "Neurons are thought to target certain desired set points (or set ranges) in the output space (i.e. function or behaviour space) corresponding to valid points or subspaces in the high-dimensional parameter space of expression levels of ion channels."

- p. 21/22: "Our computational analysis indicates that a complex high-dimensional parameter space supports the stability of neuronal excitability against perturbations that would push neurons into non-functional subspaces. The reason is that the topology of the high-dimensional space increases the likelihood of neurons returning into functional subspaces by random ion channel parameter adjustments.". The topology of this high-dimensional space is not known and it is unclear that one can use it as a reason for this behavior. What can probably be claimed is that the volume of the implied fitness functions is larger around solutions in higher dimensions, and volume is not a topological property

Reviewer #3: The authors use detailed dentate granule cell models to show that having a larger complement of ion channel types increases the robustness of these models to perturbation of channel densities and properties. This is an important point which is demonstrated convincingly in the manuscript. The substance being solid, my comments and suggestions below are mostly concerned with improving the presentation of these results.

Main point

The main result of Golowasch et al. (2002), a key paper that is cited several times in the present manuscript, is that averaging can fail due to nonlinear interactions between different types of ion channels.

In contrast, you are demonstrating success of averaging, as illustrated by the analogy between the detailed biophysical models and the toy model (= averaging) in the percentage of valid models as a function of the number of variables (Fig. 3).

It therefore seems odd that you are not mentioning or discussing the main result of Golowasch et al. (2002) explicitly. Can you explain what enables the success of averaging in your case, given that Golowasch et al. conclusively demonstrated that averaging of neural models can also fail under certain conditions? Is it simply because Golowasch et al. studied averaging across different neurons/neuron models while you are studying averaging across multiple related channel types within neurons/neuron models? Is it because your baseline models are optimized, highly valid models, and the variability you generate in the population around the baseline models respects the average represented by these valid baseline models by design? Is it because averaging indeed fails 94% to 99% of the time also in your case, if we are using your choice of uniform random sampling in the range of 0x to 2x the baseline conductance, and your definition of a valid model? Are there other possible reasons?

Minor points

1. l. 87f “large populations of stochastically generated models (Prinz et al., 2003;”: Prinz et al. (2003) introduced the databasing approach, but their database used a regular grid of parameter values, not stochastically generated ones. The references are well chosen but this sentence should be rewritten to reflect the regular grid in Prinz et al. (2003) (see also l. 95f).

2. l. 48ff “it remains unclear what role exactly the diversity of ion channel types plays regarding evolution and its contribution to functional mechanisms that impact neuronal computations.”: can you say this in plain English?

3. l. 155f “a small number of membrane time constants was sufficient to generate adequate firing patterns.”: do you really mean membrane time constants here? Usually, the membrane time constant is understood to be the passive membrane time constant tau_m = R_m * C_m.

4. The last paragraph in the Results section “The GC model cost function”, lines 117-128, refers to “The search algorithm”, lines 595-619 in Methods. This search algorithm is however used only in two specific instances in Results, namely for finding the parameters of the baseline reduced models in Fig. 1C (see also l. 617), and for finding the parameters of the baseline models with additional channel isoforms in Fig. 4 (l. 323-325; not listed in l. 617). My issue here is solely with the presentation: you should make it clear when the search algorithm is used (i.e. only to generate some baseline models) and when it is not (i.e. for what are arguably the main points of the paper, made using the “database approach” employing uniformly random scaling of channel densities and subsequent counting of valid models in Figures 2, 3A-C and 4 [after the baseline models have been established]). In restructuring those parts of Results that mention the search algorithm, you should aim to cleanly separate your use of the database approach from your use of the search algorithm to avoid confusing the reader.

5a. l. 597f “In order to search for local minima we used a conjugate gradient descent technique (Press et al., 2007).”: Please be more specific. Are you in fact not using conjugate gradient methods (which depend on the derivatives of the function to be minimized to be explicitly available) but successive line minimuzations in conjugate directions (see also l. 606ff), which works without derivatives being available? Please state the name of the minimization function you used, the source of the implementation, and the original reference for the algorithm (e.g. Brent, R.P. Algorithms for Minimization without Derivatives, Prentice Hall, Englewood Cliffs, NJ, chapter 7.)

5b. How are you dealing with the fact that some of the features in your cost function have discrete, integer values (e.g. 1. numbers of spikes fired within 200 ms under current clamp) while the minimization algorithm you use works best with smooth objective functions whose gradients are also smooth and finite?

6a. l. 623f “Each artificial channel form obtained in such a way was associated with a randomised time constant between 0x and 2x the value in the original GC model to obtain altered dynamics.”: was that range sampled uniformly? If yes, please say so.

6b. l. 191f “by randomly sampling the individual conductance densities within a range between 0x and 2x the value in the baseline model.”: was the range sampled uniformly?

7a. l. 195ff “The population of functional parameter combinations enabled us to calculate the Pearson’s correlation coefficient r for all pairs of conductance density parameters. We found weak pairwise correlations indicating low dependencies between each pair of channels and thus increasing the robustness of the model (Figure S5).”: when you say “functional parameter combinations”, do you mean the subset of GC models with valid parameter combinations? Please use consistent terminology.

7b. l. 208 “In our selection of random parameter combinations, we found suitable models covering the”: suitable models -> valid models? Please check for other instances in the manuscript where ‘valid’ should be used for consistency of terminology.

8. Figs. 3C and 3D, top right: the blue asterisks appear to be positioned too far to the left to denote the mean of the distribution.

9. l. 466 “Similar to Olypher and Calabrese (2007) and Achard and De Schutter (2006) we showed that”: shouldn’t the order of the citations be reversed based on either temporal or alphabetic order?

10. l. 529f “in which case they will most likely affect our general understanding of how evolution deals with complex organisms.”: the last sentence makes a good point, but could be improved; you could change to something like “[..] how evolution deals with complex dynamics (Koch and Laurent, 1999 [Science 284:96–98])”.

11. l. 706 “Druckmann S (2007)”: several authors missing

12. l. 727 “Gunay C, Edgerton JR, Jaeger D (2008)”: Gunay -> Günay

**Have the authors made all data and (if applicable) computational code underlying the findings in their manuscript fully available?**

Reviewer #1: Yes

Reviewer #2: Yes

Reviewer #3: Yes

PLOS authors have the option to publish the peer review history of their article (what does this mean?). If published, this will include your full peer review and any attached files.

Reviewer #1: No

Reviewer #2: **Yes: **Gaute Einevoll

Reviewer #3: No

Figure Files:

Data Requirements:

Reproducibility:

References:

---

## [Decision Letter · Decision Letter 1]

24 May 2023

Dear Mr Schneider,

We are pleased to inform you that your manuscript 'Biological complexity facilitates tuning of the neuronal parameter space' has been provisionally accepted for publication in PLOS Computational Biology.

Best regards,

Joanna Jędrzejewska-Szmek, Ph.D.

Academic Editor

PLOS Computational Biology

Daniele Marinazzo

Section Editor

PLOS Computational Biology

Reviewer's Responses to Questions

**Comments to the Authors:**

Reviewer #1: The authors have carefully replied to the revierwers and revised the manuscript accordingly.

Reviewer #2: The authors have adressed allmy queries in a satisfactory manner.

Reviewer #3: The authors have addressed all scientific issues raised during review. Some typos/nitpicking:

l. 47 “partial degeneracy(Golowasch”: space missing before the opening parenthesis

The formal definition of a valid parameter combination is given in the Methods section (lines 608ff). The authors could add “(Equation 2)” or similar at the end of the sentence “In order to become a valid parameter combination in the GC model, the error value was required to be less than two SDs away from the experimental average of each feature.” at lines 116-118.

l. 613 “sampling with successive line minimizations in conjugate directions (Brent, 2013)”.”

The quote at the end should be omitted.

l. 712 “Brent RP (2013) Algorithms for minimization without derivatives Courier Corporation.” -> Brent RP (2013) Algorithms for Minimization without Derivatives Dover Publications.

[ Dover was only temporarily owned by Courier Corporation. See also https://maths-people.anu.edu.au/~brent/pub/pub011.html ]

l. 731: 1 is the volume, the pages are 7-18. A doi link ( https://doi.org/10.3389/neuro.01.1.1.001.2007 ) could be added to help the reader locate this reference. “Gidon AA” -> Gidon A

l. 744f this reference (Goaillard & Marder) should be updated

l. 764 Ion Channels of Excitable Membranes Oxford University Press.

l. 868f this reference (Zang & Marder) should be updated, it has come out in PNAS

**Have the authors made all data and (if applicable) computational code underlying the findings in their manuscript fully available?**

Reviewer #1: Yes

Reviewer #2: None

Reviewer #3: Yes

PLOS authors have the option to publish the peer review history of their article (what does this mean?). If published, this will include your full peer review and any attached files.

Reviewer #1: No

Reviewer #2: **Yes: **Gaute Einevoll

Reviewer #3: **Yes: **Arnd Roth

---

## [Editor Report · Acceptance letter]

28 Jun 2023

PCOMPBIOL-D-23-00254R1 

Biological complexity facilitates tuning of the neuronal parameter space

Dear Dr Schneider,

I am pleased to inform you that your manuscript has been formally accepted for publication in PLOS Computational Biology. Your manuscript is now with our production department and you will be notified of the publication date in due course.

With kind regards,

Zsofia Freund
